# The Promising Effects of Erdosteine and Vitamin B in the Liver Ischemia/Reperfusion Model in Anesthetized Rats

**DOI:** 10.3390/medicina60050783

**Published:** 2024-05-08

**Authors:** Elif Eygi, Rauf Gul, Murat Aslan, Zeynel Abidin Tas, Recep Dokuyucu

**Affiliations:** 1Department of Anesthesiology and Reanimation, Gaziantep City Training and Research Hospital, Gaziantep 27470, Türkiye; aslmurat@hotmail.com; 2Department of Anesthesiology and Reanimation, School of Medicine, Gaziantep University, Gaziantep 27470, Türkiye; drraufgul@gmail.com; 3Department of Pathology, Adana City Education and Research Hospital, University of Health Sciences, Adana 01230, Türkiye; zeynelabidin46@gmail.com; 4Department of Physiology, Medical Specialization Training Center (TUSMER), Ankara 06230, Türkiye; drecepfatih@gmail.com; 5Physioclinic Private Center, Gaziantep 27560, Türkiye

**Keywords:** ischemia/reperfusion, liver, vitamin B, erdosteine, oxidative stress

## Abstract

*Background and Objectives:* Erdosteine (Erd) is an antioxidant and anti-inflammatory drug. Vitamin B has been reported to exert anti-inflammatory and antioxidant effects. In this study, we investigated the effect of erdosteine and vitamin B complex on a liver ischemia/reperfusion (I/R) model. *Materials and Methods:* Thirty-two Wistar Albino male rats weighing 350–400 g were used. The animals were randomly selected and divided into four groups. The groups are as follows: first group (Sham), second group (I/R), third group (I/R + vit B), and fourth group (I/R + vit B + Erd). Rats were subjected to 45 min of hepatic ischemia, followed by a 45 min reperfusion period in the I/R and Vitamin B + Erd groups. An amount of 150 mg/kg/day of erdosteine was given orally for 2 days, and 0.05 mL/kg of i.p. vitamin B complex was given 30 min before the reperfusion. Serum biochemical parameters were measured. Serum Total Antioxidant Status (TAS) and Total Oxidant Status (TOS) were measured, and the Oxidative Stress Index (OSI) was calculated. Hepatic tissue samples were taken for the evaluation of histopathological features. *Results*: In terms of all histopathological parameters, there were significant differences in the I/R + vit B group and I/R + vit B + Erd group compared with the I/R group (*p* < 0.01). In terms of aspartate aminotransferase (AST), alanine aminotransferase (ALT), lactate dehydrogenase (LDH), TNF-alpha, and IL-6 levels, there were significant differences between the I/R group and treatment groups (*p* < 0.01). The lowest TOS and OSI levels were obtained in the treatment groups, and these groups had statistically significantly higher TAS levels compared with the sham and I/R groups (*p* < 0.01). *Conclusions*: As a preliminary experimental study, our study suggests that these agents may have potential diagnostic and therapeutic implications for both ischemic conditions and liver-related diseases. These results suggest that the combination of vit B + Erd may be used to protect against the devastating effects of I/R injury. Our study needs to be confirmed by clinical studies with large participation.

## 1. Introduction

Liver ischemia/reperfusion (I/R) damage is primarily responsible for hepatic dysfunction or failure. Ischemic changes are seen after pringle maneuver or hemorrhagic shock during liver surgery, depending on the temporary interruption of blood flow [1].

It is known that reperfusion injury develops when blood flow returns to normal again. Oxidative phosphorylation in the cells decreases when the blood flow to the liver is cut. The energy-dependent Na^+^-K^+^-ATPase pump is inhibited in the cell membrane. Increased levels of Ca^+2^ ions in cells cause the inhibition of secondary messages and the accumulation of various proinflammatory cytokines. Ultimately, cell membrane integrity and functions are impaired [2,3]. As the cells begin to use anaerobic pathways, the accumulation of lactic acid in tissues increases and the tissue pH decreases. When the blood flow returns to ischemic liver, free oxygen radicals are formed as a result of some biochemical reactions. In this process, more damage occurs due to deterioration of antioxidant mechanisms [4,5].

Oxidative stress is the imbalance between free radicals and antioxidants in the body. Free radicals are oxygen-containing molecules with an unequal number of electrons. The unequal number of electrons causes them to react easily with other molecules. These reactions are called oxidation. Antioxidants are substances that can prevent or slow damage to cells caused by free radicals, which are unstable molecules the body produces in response to environmental and other stresses. Therefore, antioxidants protect cells against the negative effects of free radicals. Oxidative stress can cause chronic or neurodegenerative diseases, such as cancer, heart diseases, and diabetes. Free radicals, such as peroxide products and various types of ROS, cause reactions that damage proteins, lipids, and nucleic acids in the body. Various antioxidants contained in fruits and vegetables protect cell and organ systems against the negative effects of free radicals. In the literature, techniques have been developed that can measure the total oxidant level in the body, the total antioxidant level, and oxidative stress, which is a parameter that shows their balance [6,7,8]. Erdosteine is an antioxidant and anti-inflammatory drug. It inhibits the free oxygen radicals in the environment with the thiol groups it contains [6,9,10]. Vitamin B complex is a group of vitamins that are soluble in water and not stored in the body. The effect of vitamin B on ischemia/reperfusion injury in the brain, ovaries, spinal cord, and kidneys was investigated [11,12].

In this study, we investigated the effect of erdosteine and vitamin B complex on a liver ischemia/reperfusion model. Taking into account the impact of erdosteine and vitamin B, we can assess the potential of these agents in diagnosing and treating both ischemic conditions and liver-related diseases.

## 2. Materials and Methods

### 2.1. Ethics Statement and Experimental Design

Ethical approval was received from Mustafa Kemal University’s local ethics committee (approval number: 26.02.2015/2-1). In terms of the power of the study, after reviewing experimental studies similar to our study in the literature, we decided in the power analysis that the groups should consist of at least 8 rats. In this study, 32 Wistar Albino male rats weighing 350–400 g were used. It was planned that each group would consist of 8 rats. Ethics committee approval was obtained for the study. Before the experiment, all animals were kept in rooms with a 12 h at night and 12 h day circadian rhythm, the ambient temperature was 24 ± 2, and the humidity rate was 50–60%. Standard commercial pellet feed and city drinking water were used for feeding rats. The weights of rats were measured by precision weighing. The animals were randomly selected and divided into four groups [1st group (Sham), 2nd group (I/R), 3rd group (I/R + vit B), and 4th group (I/R + vit B + Erd)].

### 2.2. Surgical Procedure

Before laparotomy for anesthesia, 12 mg/kg xylazine (Rompun, Bayer, Hatay, Türkiye) and 80 mg/kg of ketamine (Ketalar, Eczacibasi, Hatay, Türkiye) were administered intraperitoneally (i.p.) in combination. Laparotomy was performed with a 3 cm incision from the midline. To create ischemia, hepatic artery and portal vein to the left and median lobe of the liver were explored and blood flow was interrupted for 45 min through the atraumatic vascular clamp. Thus, 70% of segmental (segment 2–5) and nonlethal hepatic ischemia were induced. After initiation of the ischemia process, the incision was closed by suturing. After being exposed to ischemia for 45 min, the reperfusion phase was started, and 45 min reperfusion periods were performed.

The sham group (Group 1) underwent a false surgical procedure without ischemia. The hepatic artery and portal vein were explored to the left and median lobe of the liver. Considering the duration of ischemia and reperfusion periods applied in the other groups, we waited for 45 min. A total of 45 min of ischemia and reperfusion periods was applied to the liver for the ischemia/reperfusion (I/R) group (Group 2). The I/R + vit B group (Group 3) was treated with 0.05 mL/kg i.p. vitamin B complex (Bemiks bulb Zentiva Istanbul, Türkiye) 30 min before reperfusion. The I/R + vit B + Erd group (Group 4) received 150 mg/kg/day of erdosteine orally for two days. After ischemia and 30 min before reperfusion, 0.05 mL/kg i.p. vitamin B complex was also given. The vitamin B complex we use in the treatment contains pyridoxine (B6), 20 mg niacinamide, 3 micrograms of vitamin B12, 2.5 mg folic acid, and 5 mg calcium pantothenate. Blood and liver tissue samples were collected from all animals. Blood samples were centrifuged for biochemical analysis, placed in Eppendorf tubes, and stored at −20 °C until the day of measurement. Tissue samples were placed in a 10% formalin solution for histopathological analysis and stored at −80 °C until the day of examination.

### 2.3. Histopathological Investigations

For histopathological analysis, 2 slices with 3 mm of thickness perpendicular to the long axis were taken from each liver tissue and placed in 10% formalin solution. They were embedded in paraffin using routine histological methods. Microtome (Leica Rotary) sections were taken from the tissues embedded in paraffin blocks, the sections were stained with Hematoxylin–Eosin (H&E), and the standard protocol was applied. The preparations were examined by a specialized histopathologist at ×100 magnification in a light microscope (Olympus Clinical Microscope BX45, Tokyo, Japan). Histopathological examination of the liver tissue samples was taken into account, and the overall tissue integrity I/R damage was scored as semiquantitative [13,14]. In the pathological examination of the liver tissue samples, similar scoring studies were taken as reference in previous liver I/R model studies [13,14]. All liver tissue samples were evaluated for cellular swelling, lipoid degeneration, sinusoidal congestion, hemorrhage, inflammatory cell infiltration, lobular necrosis, and apoptosis [15].

### 2.4. Biochemical Investigations

Blood samples taken by intracardiac method were kept in biochemistry tubes for 25–30 min and then they were centrifuged at 4000 rpm for 10 min at +4 °C. Serum samples were placed in Eppendorf tubes for biochemical analysis and first kept at −20 °C and then stored at −80 °C until measurement day. Serum AST, ALT, and LDH activities were analyzed in an autoanalyzer (Architect c8000, Clinical Chemistry Analyzer, Abbott, IL, USA). TNF-alpha and IL-6 concentrations were determined by enzyme-linked immunosorbent assay (ELISA) kits (Biosource Invitrogen Immunoassay, Catalog no: KRC3011, Wilmington, NC, USA; Biosource Invitrogen Immunoassay, Catalog no: KRC0012, Wilmington, NC, USA). IL-6 and TNF-alpha levels were expressed in pg/mL.

TAS (mmol/L) levels were measured colorimetrically using a commercially purchased kit (Relassay, Gaziantep, Türkiye). The new automated method is based on the bleaching of the characteristic color of a more stable ABTS (2,2′-Azino-bis (3-ethylbenzothiazoline-6-sulfonic acid) radical cation by antioxidants. The assay has excellent precision values of less than 3%. Results are expressed as mmol Trolox equivalent/L [8].

TOS (µmol/L) levels were measured colorimetrically using a commercially available kit (Relassay, Türkiye). In the method used, oxidants present in the sample oxidize the iron ion-o-dianisidine complex to iron ion. The oxidation reaction is enhanced by glycerol molecules abundant in the reaction medium. The ferric ion produces a colored complex with xylenol orange in an acidic environment. Color intensity, which can be measured spectrophotometrically, is related to the total amount of oxidant molecules present in the sample. The test is calibrated with hydrogen peroxide and results are expressed in micromolar hydrogen peroxide equivalents per liter (μmol H_2_O_2_ equivalent/L) [7].

Automated measurement techniques were used to determine serum TAS, TOS, and OSI. OSI was defined as the ratio of TOS to TAS. Specifically, OSI (arbitrary unit) = TOS (mmol H_2_O_2_ Eq/L)/TAS (mmol Trolox Eq/L) [9].

### 2.5. Statistical Analysis

All statistical evaluations were conducted using the Social Sciences Statistical Package Version 27.0 (SPSS Comp, Chicago, IL, USA). Continuous variables were represented as mean ± standard deviation. To evaluate the disparities among numerical variables, we employed the Kruskal–Wallis test. The Mann–Whitney U-test was used for post hoc analysis. *p*-value below 0.05 was deemed statistically significant.

## 3. Results

### 3.1. Histopathological Findings

In terms of histopathological parameters, there was a significant difference between the sham and I/R group in terms of cellular swelling, sinusoidal congestion, and inflammatory cell infiltration (*p* < 0.01). In addition, in terms of all histopathological parameters, there was also a significant difference in the I/R + vit B group and I/R + vit B + Erd group compared with the I/R group (*p* < 0.01). According to the histopathological parameters between the I/R + Vit B group and I/R + vit B + Erd group, there was a significant difference in terms of cellular swelling, sinusoidal congestion, and inflammatory cell infiltration, but no difference in apoptosis (Table 1). In the microscopic examination of the sham group, the liver tissue was normal (Figure 1a). Compared with the sham group, significant cellular swelling, sinusoidal congestion, and inflammatory cell infiltration were observed in the I/R group (Figure 1b). Inflammatory cells, congestion, cellular swelling, and apoptosis were significantly decreased in both treatment groups (Figure 1c,d).

### 3.2. Biochemical Examination Findings

In the analysis of blood samples, in terms of AST, ALT, LDH, TNF-alpha, and IL-6 levels, there was a significant difference between the sham group and the I/R group (*p* < 0.001). In terms of AST, LDH, TNF-alpha, and IL-6 levels, except for ALT, there was a significant difference between the I/R + vit B group and the I/R group (*p* < 0.001). In terms of all the above parameters, there was a significant difference between the I/R group and the I/R + vit B + Erd group (*p* < 0.001). In terms of TNF-alpha and IL-6 levels, there was a significant difference between the I/R + vit B group and the I/R + vit B + Erd group (*p* < 0.01, *p* < 0.05, respectively) (Table 2).

A comparison of the groups in terms of oxidative stress parameters is shown in Table 3. In terms of TOS, there was a significant difference between the sham and I/R groups (*p* < 0.001). There was a significant difference between the treatment groups (I/R + vit B and I/R + vit B + Erd) and I/R group (*p* < 0.05, *p* < 0.01, respectively). In terms of OSI, there was a significant difference between the sham and I/R groups (*p* < 0.001). There was a significant difference between the treatment groups (I/R + vit B and I/R + vit B + Erd) and I/R group (*p* < 0.05, *p* < 0.01, respectively) (Table 3).

## 4. Discussion

In our study, which was a preliminary experimental study, we investigated a pioneering study to examine the effect of erdosteine and vitamin B complex on the liver ischemia/reperfusion rat model. Histopathological examinations revealed significant cellular swelling, sinusoidal congestion, and inflammatory cell infiltration in I/R injury. We found that the vitamin B complex and erdosteine we used in our study had a protective and corrective effect on this damage. However, it was determined that vitamin B and erdosteine had protective effects in terms of biochemical parameters. Especially in terms of OSI values, a study in the literature found that an antioxidant-effective substance administered in hepatic I/R injury reduced oxidative stress in organs [16].

In recent years, many studies on erdosteine have mentioned the protective role of erdosteine in various tissue injuries mediated by oxidative stress products and inflammatory factors. Erdosteine prevents the accumulation of free oxygen radicals when the production accelerates and strengthens antioxidant cellular protective mechanisms. The end result is a tissue-protective effect that reduces lipid peroxidation, neutrophil infiltration, and cellular apoptosis mediated by harmful agents. It has been reported that erdosteine and vitamin B complex prevent the accumulation of free oxygen radicals and increase antioxidant cellular protective mechanisms [10,17,18,19]. In our study, we observed the protective effects of erdosteine and vitamin B complex on biochemical parameters, oxidative stress, and histopathological parameters in the I/R model on rats as a preliminary experimental study.

In a study on biliary obstruction liver fibrosis in rats, antioxidant and antifibrotic effects of 10 mg/kg erdosteine were investigated [15]. Plasma cytokines, AST, ALT, and LDH were reported to be significantly reduced by erdosteine treatment for 28 days. In the study, conducted by Şener et al., in terms of biochemical parameters (AST, ALT, LDH, TNF-alpha, and IL-6 values), it was seen that there were significant differences between the I/R and erdosteine groups [15]. Unlike that study, in our study, we gave 150 mg/kg/day of erdosteine orally for 2 days, and 0.05 mL/kg i.p. vitamin B complex was given 30 min before the reperfusion. In terms of biochemical parameters (AST, ALT, LDH, TNF-alpha, and IL-6 values), there were significant differences between the I/R group and treatment groups.

The effects of erdosteine and substances with different protective effects on ischemia/reperfusion (I/R) injury have been documented in various organ systems, including the heart, kidneys, central nervous system, lungs, skeletal muscles, and intestines [20,21,22,23]. Cao et al. [20] reported that erdosteine has a protective effect which reduces oxidative stress and neutrophil accumulation against distant organ lung injury after hindlimb I/R. In a separate study, researchers reported the potential protective effects of erdosteine, an antioxidant, on unilateral testicular reperfusion injury in rats. The study involved four groups of rats: control, torsion, torsion/detorsion, and torsion/detorsion + erdosteine [20]. Tunç et al. [22] investigated whether rats treated with erdosteine and ebselen displayed a beneficial effect against intestinal injury and their combinations. Both erdosteine and ebselen have been reported to reduce intestinal I/R damage. Ozer et al. [23] reported that ischemia/reperfusion has a negative effect on erythrocyte deformity and that it causes deterioration in blood flow and tissue perfusion in infrarenal rat aorta. Erdosteine has been shown to have beneficial effects by reversing the undesirable effects of ischemia/reperfusion. Barlas et al. [21] reported that erdosteine has positive effects on the harmful effects of experimental hepatic ischemia/reperfusion injury. The erdosteine treatment group improved histopathological abnormalities compared with the control group. It was also found that this treatment significantly reduced serum liver function test values. Erdosteine was reported to improve oxidative stress in treatment groups. They provided ischemia for 60 min and reperfusion for 90 min. An amount of 100 mg/kg of erdosteine was administered 2 h before ischemia induction. In the preoperative treatment group, 100 mg/kg/day of erdosteine was administered daily for 10 days before the operation and 2 h before ischemia induction [21]. Unlike Barlas’s study, in our study, we provided ischemia for 45 min and reperfusion for 45 min. We gave 150 mg/kg/day of erdosteine orally for 2 days and 0.05 mL/kg i.p. vitamin B complex 30 min before the reperfusion. In our study, erdosteine was found to improve histopathological and biochemical abnormalities.

There are some studies in the literature regarding the effects of vitamin B types [24,25,26]. Sanches et al. [24] showed that riboflavin has anti-inflammatory and antioxidant properties in experimental sepsis and ischemia/reperfusion injury. Riboflavin has shown antioxidant and anti-inflammatory effects in ischemic liver and protects hepatocytes against I/R damage. In another study, Huang et al. [26] investigated the effects of folic acid and B_6_ and B_12_ vitamins on plasma homocysteine, as well as learning and memory functions. Folic acid intake not only reduced plasma homocysteine concentration but also stimulated the recovery of learning and memory functions of rats with cerebral ischemia. Vitamin B_6_ and vitamin B_12_ together, with the effect of folic acid on cerebral ischemia rats, have been reported to be positive. Folic acid pretreatment has blunted myocardial dysfunction during ischemia and has been reported to improve postreperfusion injury. Hamed et al. [25] showed that pretreatment with high-dose oral folic acid significantly reduced ischemic dysfunction during coronary occlusion, increased function following reperfusion, and reduced infarct size. Unlike other studies in the literature, in our study, the vitamin B complex we used in the treatment contains pyridoxine (B_6_), 20 mg niacinamide, 3 micrograms of vitamin B_12_, 2.5 mg folic acid, and 5 mg calcium pantothenate. We showed that vitamin B complex contributes to the protective effects of erdosteine and reduces I/R damage.

Considering the evidence from studies that highlight the beneficial impact of erdosteine and vitamin B on ischemia/reperfusion injury, the findings of this particular investigation indicate a reduction in the harmful effects of such injury on the liver. When erdosteine and vitamin B treatment groups were compared with the I/R group, positive changes were observed in histopathological, biochemical, and oxidative stress parameters. In the context of this study, the treatment groups exhibited a reduction in inflammatory cell infiltration. Additionally, this treatment led to a significant decrease in serum liver function test values. However, when oxidative stress parameters were evaluated, it was determined that both erdosteine and vitamin B improved oxidative stress parameters.

## 5. Conclusions

In summary, as a preliminary experimental study, this research represents a pioneering effort in examining the impact of erdosteine and vitamin B complex on liver ischemia/reperfusion rat model. Our study suggests that these agents may have potential diagnostic and therapeutic implications for both ischemic conditions and liver-related diseases. Although the I/R + vit B + Erd group showed better results than the I/R + vit B group, positive effects were also seen in the I/R + vit B group. These results suggest that the combination of vit B + Erd may be used to protect against the devastating effects of I/R injury. Our study needs to be confirmed by clinical studies with large participation.

## Figures and Tables

**Figure 1 medicina-60-00783-f001:**
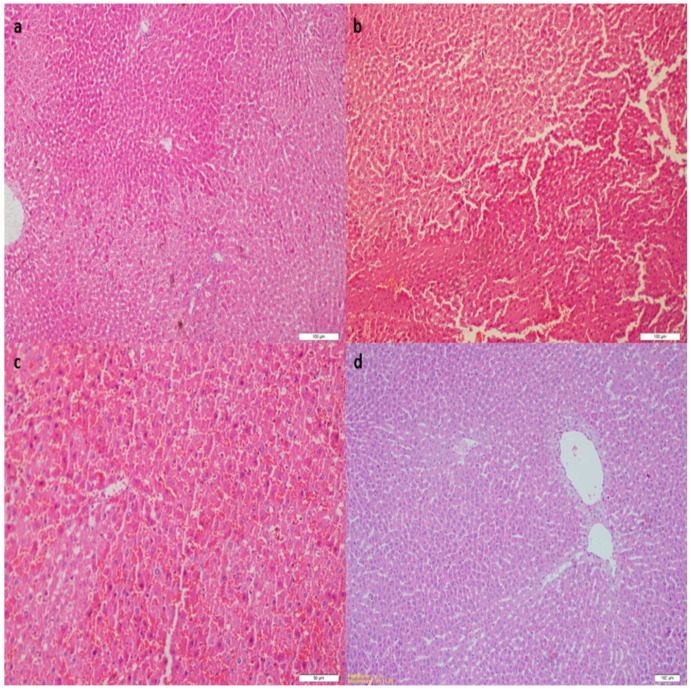
Histopathological determinations of the groups: The liver tissue was normal in Group 1 (**a**). Significant cellular swelling, sinusoidal congestion, and inflammatory cell infiltration were observed in Group 2 (**b**). Inflammatory cells, congestion, cellular swelling, and apoptosis were significantly decreased in Groups 3 and 4 (**c**,**d**).

**Table 1 medicina-60-00783-t001:** Statistical comparison of the scoring results of the histopathological parameters in groups (Mean ± SE).

	Group 1 (Sham)	Group 2 (I/R)	Group 3 (I/R + vit B)	Group 4 (I/R + vit B + Erd)
Cellular Swelling	1.43 ± 0.53	3.13 ± 0.64 ^a^**	2.00 ± 0.00 ^b^**	1.29 ± 0.49 ^b,c^***
Steatosis/lipoid degeneration	0.00 ± 0.00	0.00 ± 0.00	0.00 ± 0.00	0.00 ± 0.00
Sinusoidal congestion	2.14 ± 0.69	3.88 ± 0.35 ^a^**	3.62 ± 0.52 ^b^**	2.29 ± 0.95 ^b,c^**
Hemorrhage	0.00 ± 0.00	0.00 ± 0.00	0.00 ± 0.00	0.00 ± 0.00
Inflammatory cell	1.14 ± 0.38	2.00 ± 0.76 ^a^**	1.12 ± 0.35 ^b^*	1.00 ± 1.00 ^b,c^**
Lobular necrosis	0.00 ± 0.00	0.00 ± 0.00	0.00 ± 0.00	0.00 ± 0.00
Apoptosis	0.14 ± 0.38	0.63 ± 0.74	0.25 ± 0.46 ^b^**	0.29 ± 0.49 ^a,b^**

^a^: vs. Sham; ^b^: vs. I/R; ^c^: vs. Erd + I/R; *: *p* < 0.05; **: *p* < 0.01; ***: *p* < 0.001.

**Table 2 medicina-60-00783-t002:** Statistical comparison of the biochemical parameters in groups (Mean ± SE).

	Sham	I/R	I/R + vit B	I/R + vit B + Erd
AST (U/L)	116.1 ± 9.7	1436.0 ± 127.4 ^a^***	799.40 ± 71.1 ^b^***	676.90 ± 51.0 ^b^***
ALT (U/L)	42.57 ± 4.6	655.60 ± 54.3 ^a^***	528.10 ± 34.3	405.0 ± 35.5 ^b^***
LDH (mg/dL)	302.30 ± 37.4	1428.0 ± 107.5 ^a^***	955.20 ± 51.8 ^b^***	801.30 ± 36.1 ^b^***
TNF-alfa (ng/mL)	59.89 ± 2.2	158.30 ± 6.2 ^a^***	104.40 ± 5.5 ^b^***	77.11 ± 6.1 ^b^***^, c^**
IL-6 (ng/mL)	51.75 ± 2.6	110.60 ± 12.81 ^a^***	74.63 ± 5.0 ^b^**	60.25 ± 4.2 ^b^***^, c^*

^a^: Sham vs. I/R; ^b^: vs. I/R; **^c^**: IR+ vit B + Erd vs. Erd + I/R. *: *p* < 0.05; **: *p* < 0.01; ***: *p* < 0.001.

**Table 3 medicina-60-00783-t003:** Comparison of groups in terms of oxidative stress parameters (Mean ± SE).

	Sham	I/R	I/R + vit B	I/R + vit B + Erd
TAS (mmol/L)	1.096 ± 0.22	0.713 ± 0.03	0.856 ± 0.06	0.710 ± 0.06
TOS (µmol/L)	25.70 ± 5.0	104.80 ± 12.4 ^a^***	65.87 ± 12.3 ^b^*	47.69 ± 4.4 ^b^**
OSI (TOS/TAS)	31.01 ± 10.0	145.70 ± 15.7 ^a^***	78.38 ± 13.9 ^b^**	72.18 ± 11.8 ^b^**

^a^: Sham vs. I/R. ^b^: vs. I/R. *: *p* < 0.05. **: *p* < 0.01. ***: *p* < 0.001.

## Data Availability

The data that support the findings of this study are available from the corresponding author upon reasonable request.

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
