# Peer review of "The Promising Effects of Erdosteine and Vitamin B in the Liver Ischemia/Reperfusion Model in Anesthetized Rats"

_medicina, 2024, doi:10.3390/medicina60050783_

Round 1

Reviewer 1 Report

Comments and Suggestions for Authors

1. Originality and Relevance: The study addresses a relevant question concerning the protective effects of Erdosteine and Vitamin B complex in a liver ischemia/reperfusion (I/R) model using Wistar Albino rats. This research taps into the vital area of potential treatments for liver damage related to surgical interventions, with a focus on oxidative stress and inflammation—topics of high relevance in medical research.

2. Methodology: The experimental design and methodologies used, including ethical considerations and detailed procedural descriptions, adhere well to standard practices in biomedical research. However, there are areas where the methodology could be significantly improved:

  • Sample Size Justification: There is no mention of sample size calculation. It is important to establish that the study has adequate power to detect a significant effect.

3. Results: The results section is detailed with clear statistical analysis, demonstrating significant improvements in liver function and reductions in damage markers with treatment. However:

  • Data Presentation: The presentation of data in tables is clear, but the manuscript could benefit from additional graphical representations of key data points to enhance readability and interpretation.
  • Consistency in Results: There are inconsistencies in the text regarding the comparisons between groups, particularly in oxidative stress parameters. Clarifications and possibly additional statistical analysis could resolve these issues.

4. Discussion and Interpretation: The discussion outlines the implications of the findings and situates them within the broader context of existing literature effectively. Nonetheless, it overreaches somewhat in the conclusions about the clinical applicability of the treatments without sufficient evidence from more extensive or diverse studies.

5. Language and Style: The manuscript is generally well-written but could benefit from proofreading to correct minor grammatical errors and improve clarity in some sections. The use of technical terms is appropriate, and the style is consistent with scientific discourse.

6. Figures and Tables: All figures and tables are relevant and support the textual content. However, enhancing figure quality for better visual impact and including additional charts summarizing the main findings could improve the manuscript’s communicative effectiveness.

7. Ethical Considerations: The study meets ethical standards with proper committee approvals mentioned. This aspect of the research is well-handled and adds credibility to the work.

Conclusion: This manuscript presents interesting findings on the protective effects of Erdosteine and Vitamin B in liver I/R injury. However, to strengthen the study's validity and address the minor issues noted regarding methodology and data presentation, I recommend accepting this manuscript with minor corrections.

Comments on the Quality of English Language

The manuscript is generally well-written but could benefit from proofreading to correct minor grammatical errors and improve clarity in some sections. The use of technical terms is appropriate, and the style is consistent with scientific discourse.

Author Response

Dear Editor,

Thank you for your interest in our manuscript; we are sending its revised form after carefully considering the comments of the Reviewer’s and having modified it accordingly. We have made a point-by-point response to the Reviewer’s comments. The changes are coloured in the text and are summarized below. We look forward to hearing from you soon.

Yours faithfully,

REVIEWERS’ COMMENT:

Reviewer 1:

  1. Originality and Relevance: The study addresses a relevant question concerning the protective effects of Erdosteine and Vitamin B complex in a liver ischemia/reperfusion (I/R) model using Wistar Albino rats. This research taps into the vital area of potential treatments for liver damage related to surgical interventions, with a focus on oxidative stress and inflammation—topics of high relevance in medical research.

  1. Methodology: The experimental design and methodologies used, including ethical considerations and detailed procedural descriptions, adhere well to standard practices in biomedical research. However, there are areas where the methodology could be significantly improved:

Sample Size Justification: There is no mention of sample size calculation. It is important to establish that the study has adequate power to detect a significant effect.

R: Since our study was an experimental study on rats, we created each group with 8 rats since the sample size was required to be at least 8 in terms of the power of the study. After reviewing experimental studies similar to our study in the literature, we decided in the power analysis that the groups should consist of at least 8 rats. (https://pubmed.ncbi.nlm.nih.gov/32685084 ; https://pubmed.ncbi.nlm.nih.gov/37606113/ ; https://pubmed.ncbi.nlm.nih.gov/36404629/  )

In the Materials and Methods section, “In terms of the power of the study, after reviewing experimental studies similar to our study in the literature, we decided in the power analysis that the groups should consist of at least 8 rats.”.

  1. Results: The results section is detailed with clear statistical analysis, demonstrating significant improvements in liver function and reductions in damage markers with treatment. However:

Data Presentation: The presentation of data in tables is clear, but the manuscript could benefit from additional graphical representations of key data points to enhance readability and interpretation.

Consistency in Results: There are inconsistencies in the text regarding the comparisons between groups, particularly in oxidative stress parameters. Clarifications and possibly additional statistical analysis could resolve these issues.

R: Thank you for your suggestions. Oxidative stress parameters in the text were edited.

“Comparison of groups in terms of oxidative stress parameters were showed in table 3. In terms of TAS, there was no significant difference between the groups in the evaluation of TAS (p>0.05). In terms of TOS, there was a significant difference between the Sham and I/R (p<0.001). There was a significant difference between the treatment groups (I/R + VIT B and I/R + VIT B + ERD) and I/R group (P<0.05, p<0.01, respectively). In terms of OSI, there was a significant difference between the Sham and I/R (p<0.001). There was a significant difference between the treatment groups (I/R + VIT B and I/R + VIT B + ERD) and I/R group (P<0.05, p<0.01, respectively) (Table 3).”

  1. Discussion and Interpretation: The discussion outlines the implications of the findings and situates them within the broader context of existing literature effectively. Nonetheless, it overreaches somewhat in the conclusions about the clinical applicability of the treatments without sufficient evidence from more extensive or diverse studies.

  1. Language and Style: The manuscript is generally well-written but could benefit from proofreading to correct minor grammatical errors and improve clarity in some sections. The use of technical terms is appropriate, and the style is consistent with scientific discourse.

  1. Figures and Tables: All figures and tables are relevant and support the textual content. However, enhancing figure quality for better visual impact and including additional charts summarizing the main findings could improve the manuscript’s communicative effectiveness.

R: Thank you for your suggestions. A Graphical Abstract was added for To facilitate the reader's understanding.

  1. Ethical Considerations: The study meets ethical standards with proper committee approvals mentioned. This aspect of the research is well-handled and adds credibility to the work.

Conclusion: This manuscript presents interesting findings on the protective effects of Erdosteine and Vitamin B in liver I/R injury. However, to strengthen the study's validity and address the minor issues noted regarding methodology and data presentation, I recommend accepting this manuscript with minor corrections.

Comments on the Quality of English Language

The manuscript is generally well-written but could benefit from proofreading to correct minor grammatical errors and improve clarity in some sections. The use of technical terms is appropriate, and the style is consistent with scientific discourse.

Reviewer 2 Report

Comments and Suggestions for Authors

Typically, rather than framing the title with the primary question of the paper, starting with the principal answer or conclusion can be more impactful, in my opinion. However, initiating the article with a question is also perfectly acceptable.

Overall, the paper is concise and gets straight to the point, making it easily understandable, even for those not well-versed in the liver research field. Nevertheless, several grammatical errors need to be addressed. Please make sure that every period, space, and uppercase letter is correctly placed.

Regarding novelty: In reference 19, you cite an article by Barlas et al., and upon reading it, I noticed that it appears to be a very similar study to the one you are presenting. In the discussion section, you reference this study, but it is not entirely clear what the differences are between your study and theirs. One notable distinction I quickly identified is the addition of vitamin B to the treatment groups; however, it is important to ensure that these differences are clear.

In terms of clinical importance: While it is understandable that this study is translational/experimental, it might be beneficial for readers to understand if there are any intentions to assess the impact of the treatment in humans. Are there any ongoing clinical trials or literature discussing early-phase clinical trials in development? Additionally, where do you envision your findings leading in the future?

Discussion: The discussion section needs to be improved. One suggestion is to divide the discussion into "key learning points" for you and then compare these findings with what has been previously reported in the literature. It appears that in the discussion, each paragraph corresponds to a similar study that has been conducted before, which is not aligned with the overall purpose of the discussion.

In summary, while the information presented is concise, engaging, and covers an impactful topic, there is room for improvement in how the information is communicated, particularly in the discussion section.

Comments on the Quality of English Language

It could be a good idea that a native English speaker can revise how the information is presented. 

Author Response

Dear Editor,

Thank you for your interest in our manuscript; we are sending its revised form after carefully considering the comments of the Reviewer’s and having modified it accordingly. We have made a point-by-point response to the Reviewer’s comments. The changes are coloured in the text and are summarized below. We look forward to hearing from you soon.

Yours faithfully,

REVIEWERS’ COMMENT:

Reviewer 2:

  1. Typically, rather than framing the title with the primary question of the paper, starting with the principal answer or conclusion can be more impactful, in my opinion. However, initiating the article with a question is also perfectly acceptable.

R: We change the title as “The Promising Effects of Erdosteine and Vitamin B in the Liver Ischemia/Reperfusion Model in Rats

Overall, the paper is concise and gets straight to the point, making it easily understandable, even for those not well-versed in the liver research field. Nevertheless, several grammatical errors need to be addressed. Please make sure that every period, space, and uppercase letter is correctly placed.

R: Thank you for your suggestions. Language, spelling and grammatical errors were edited.

Regarding novelty: In reference 19, you cite an article by Barlas et al., and upon reading it, I noticed that it appears to be a very similar study to the one you are presenting. In the discussion section, you reference this study, but it is not entirely clear what the differences are between your study and theirs. One notable distinction I quickly identified is the addition of vitamin B to the treatment groups; however, it is important to ensure that these differences are clear.

R: Thank you for your suggestions. We revised that paragraph as” Barlas et al [21] reported that Erdosteine has positive effects on the harmful effects of the experimental hepatic ischemia-reperfusion injury. Erdosteine treatment group improved histopathological abnormalities compared to the control group. It was also found that this treatment significantly reduced serum liver function test values. Erdosteine was reported to improve oxidative stress in treatment groups. They provided ischemia for 60 minutes and reperfusion for 90 minutes. 100 mg/kg erdosteine was administered 2 hours before ischemia induction. In the preoperative treatment group, 100 mg/kg/day erdosteine was administered daily for 10 days before the operation and 2 hours before ischemia induction [21]. Unlike Barlas's study, in our study, we provided ischemia for 45 minutes and reperfusion for 45 minutes. We gave 150 mg/kg/day orally erdosteine for two days and 30 min after reperfusion and 30 min after reperfusion before 0.05 ml / kg I.p. Vitamin B complex.”

In terms of clinical importance: While it is understandable that this study is translational/experimental, it might be beneficial for readers to understand if there are any intentions to assess the impact of the treatment in humans. Are there any ongoing clinical trials or literature discussing early-phase clinical trials in development? Additionally, where do you envision your findings leading in the future?

R: We revised the discussion and conclusion paragraph as “In our study, as a preliminary experimental study, we investigated a pioneering study to examine the effect of erdosteine and vitamin B complex on the liver ischemia/reperfusion rat model.”   “Our study suggests that these agents may have potential diagnostic and therapeutic implications for both ischemic conditions and liver-related diseases. Although the IR + Vit B + ERD group showed better results than the IR + Vit B group, positive effects were also seen in the IR + Vit B group. These results suggest that the combination of Vit B + ERD may be used to protect against the devastating effects of I/R injury. Our study is important as it is a pioneering study in clinical studies. Our study needs to be confirmed by clinical studies with large participation.”

Discussion: The discussion section needs to be improved. One suggestion is to divide the discussion into "key learning points" for you and then compare these findings with what has been previously reported in the literature. It appears that in the discussion, each paragraph corresponds to a similar study that has been conducted before, which is not aligned with the overall purpose of the discussion.

R: Discussion section was edited.

In summary, while the information presented is concise, engaging, and covers an impactful topic, there is room for improvement in how the information is communicated, particularly in the discussion section.

R: Discussion section was edited.

It could be a good idea that a native English speaker can revise how the information is presented.

R: Language, spelling and grammatical errors were edited.

Reviewer 3 Report

Comments and Suggestions for Authors

The authors evaluated the effects of erdosteine and vitamin B on liver ischemic/reperfusion injury in rats. While the idea is interesting and the authors have shown positive effects of the mentioned drugs on I/R injury, the Manuscript is inadequately written with a great number of necessary corrections, especially regarding the materials and methods section. Results should be presented in a more orderly manner. The discussion should be more convincing. 

These points require correction or extension:

1.     The introduction should include a brief explanation of TAS, TOS, and OSI parameters, otherwise the reader can not adequately understand the results.

2.     Materials and methods: how was the randomization conducted? How many rats were in each group?

3.     Materials and methods: Authors mention the semiquantitative histopathological scoring but do not name the scoring, and the given reference does not provide data about the scoring.

4.     Materials and methods: Authors state that 'Automated measurement techniques were used to determine serum TAS, TOS, and OSI’. However, no precise information on these techniques and devices used is given, while the mentioned markers seem to be the primary objectives of the study.

5.     The discussion should focus on the effects of vitamin B and not folic acid (it is not the subject of research). Are there any studies about the beneficial effects of erdosteine and vitamin B on humans?

6.     Authors should discuss the potential clinical benefits of their study.

7.     Lines 188- 191 and 192-195- please use the proper references (instead of (11)).

8.     The tables are complicated to read. Authors should make one Table with the mean values of obtained parameters; and use other Tables to present statistical results properly. It is not even mentioned in the tables which statistical test was used. The authors also mentioned the Spearman correlation that is not visible in the Results. 

9. 

Comments on the Quality of English Language

    The comprehensive correction of the English language by a native speaker is mandatory.

Author Response

Dear Editor,

Thank you for your interest in our manuscript; we are sending its revised form after carefully considering the comments of the Reviewer’s and having modified it accordingly. We have made a point-by-point response to the Reviewer’s comments. The changes are coloured in the text and are summarized below. We look forward to hearing from you soon.

Yours faithfully,

REVIEWERS’ COMMENT:

Reviewer 3:

The authors evaluated the effects of erdosteine and vitamin B on liver ischemic/reperfusion injury in rats. While the idea is interesting and the authors have shown positive effects of the mentioned drugs on I/R injury, the Manuscript is inadequately written with a great number of necessary corrections, especially regarding the materials and methods section. Results should be presented in a more orderly manner. The discussion should be more convincing.

These points require correction or extension:

  1. The introduction should include a brief explanation of TAS, TOS, and OSI parameters, otherwise the reader can not adequately understand the results.

R: The introduction was revised. “Oxidative stress is the imbalance between free radicals and antioxidants in the body. Free radicals are oxygen-containing molecules with an unequal number of electrons. The unequal number of electrons causes them to react easily with other molecules. These reactions are called oxidation. Antioxidants are substances that can prevent or slow damage to cells caused by free radicals, which are unstable molecules the body produces in response to environmental and other stresses. Therefore, antioxidants protect cells against the negative effects of free radicals. Oxidative stress can cause chronic or neurodegenerative diseases such as cancer, heart diseases and diabetes. Free radicals, such as peroxide products and various types of ROS, cause reactions that damage proteins, lipids and nucleic acids in the body. Various antioxidants contained in fruits and vegetables protect cell and organ systems against the negative effects of free radicals. In the literature, techniques have been developed that can measure the total oxidant level in the body, the total antioxidant level and oxidative stress, which is a parameter that shows their balance [6-8].”

  1. Materials and methods: how was the randomization conducted? How many rats were in each group?

R: Materials and methods was edited. “In terms of the power of the study, after reviewing experimental studies similar to our study in the literature, we decided in the power analysis that the groups should consist of at least 8 rats. In the study, 32 Wistar Albino male rats weighing 350 - 400 g were used.  It was planned that each group would consist of 8 rats.”

  1. Materials and methods: Authors mention the semiquantitative histopathological scoring but do not name the scoring, and the given reference does not provide data about the scoring.

R: References were added as “Histopathological examination of the liver tissue samples taken into account the overall tissue integrity I / R damage was scored as semiquantitative [13,14]. In the pathological examination of the liver tissue samples, similar scoring studies were taken as reference in previous liver I / R model studies [13,14].”

  1. Materials and methods: Authors state that 'Automated measurement techniques were used to determine serum TAS, TOS, and OSI’. However, no precise information on these techniques and devices used is given, while the mentioned markers seem to be the primary objectives of the study.

R: Materials and methods were edited as “TAS (mmol/L) levels were measured colorimetrically using a commercially purchased kit (Relassay, Turkey). The new automated method is based on the bleaching of the characteristic color of a more stable ABTS (2,2′ - Azino-bis(3-ethylbenzothiazoline-6-sulfonic acid)) radical cation by antioxidants. The assay has excellent precision values of less than 3%. Results are expressed as mmol Trolox equivalent/L [8].

“TOS (µmol/L) levels were measured colorimetrically using a commercially available kit (Relassay, Turkey). In the method used, oxidants present in the sample oxidize the iron ion-o-dianisidine complex to iron ion. The oxidation reaction is enhanced by glycerol molecules abundant in the reaction medium. The ferric ion produces a colored complex with xylenol orange in an acidic environment. Color intensity, which can be measured spectrophotometrically, is related to the total amount of oxidant molecules present in the sample. The test is calibrated with hydrogen peroxide and results are expressed in micromolar hydrogen peroxide equivalents per liter (μmol H2O2 equivalent/L) [7].”

  1. The discussion should focus on the effects of vitamin B and not folic acid (it is not the subject of research). Are there any studies about the beneficial effects of erdosteine and vitamin B on humans?

R: Thank you for your suggestions. Discussion section was edited.

  1. Authors should discuss the potential clinical benefits of their study.

R: Thank you for your suggestions. Discussion section was edited.

  1. Lines 188- 191 and 192-195- please use the proper references (instead of (11)).

R: Thank you for your suggestion. The paragraph was revised as “The effects of erdosteine and substances with different protective effects on Ischemia-reperfusion (I/R) injury have been documented in various organ systems, including the heart, kidneys, central nervous system, lungs, skeletal muscles and intestines [20-23]. Cao et al [20] was reported that erdosteine has a protective effect which reduces oxidative stress and neutrophil accumulation against distant organ lung injury after hindlimb ischemia-reperfusion (I/R). In a separate investigation, researchers explored the potential protective effects of erdosteine, an antioxidant, on unilateral testicular reperfusion injury in rats. The study involved four groups of rats: control, torsion, torsion/detorsion, and torsion/detorsion+erdosteine [20]. Tunç et al [22] was investigated whether rats treated with erdosteine and ebselen had a beneficial effect against intestinal injury and their combinations. Both erdosteine and ebselen have been reported to reduce intestinal I/R damage. Ozer et al [23] have been reported that Ischemia/reperfusion has a negative effect on erythrocyte deformity and it causes deterioration in blood flow and tissue perfusion in infrarenal rat aorta. Erdosteine has been shown to have beneficial effects by reversing the undesirable effects of ischemia/reperfusion. Barlas et al [21]….”

  1. The tables are complicated to read. Authors should make one Table with the mean values of obtained parameters; and use other Tables to present statistical results properly. It is not even mentioned in the tables which statistical test was used. The authors also mentioned the Spearman correlation that is not visible in the Results.

R: Results section was revised and Spearman correlation wasn’t mentioned in our study.

  1. The comprehensive correction of the English language by a native speaker is mandatory.

R: Thank you for your suggestions. Language, spelling and grammatical errors were edited.

Round 2

Reviewer 3 Report

Comments and Suggestions for Authors

The authors have revised the Manuscript properly, however there are still several points to be corrected:

1. The results presented in Tables should not be repeated in text. Only the most important ones should be emphasized.

2. Authors state in their comments that 'Spearman correlation wasn’t mentioned in our study', however Lines 156 and 157 in the Manuscript state: ' Spearman’s correlation analysis was used to evaluate correlation tests between numerical variables'.

Comments on the Quality of English Language

Quality of English language remains improper. The Manuscript must be revised by the native speaker. 

Author Response

Dear Editor,

Thank you for your interest in our manuscript; we are sending its revised form after carefully considering the comments of the Reviewer’s and having modified it accordingly. We have made a point-by-point response to the Reviewer’s comments. The changes are colored in the text and are summarized below. We look forward to hearing from you soon.

Yours faithfully,

REVIEWERS’ COMMENT:

Reviewer 3:

The authors have revised the Manuscript properly, however there are still several points to be corrected:

  1. The results presented in Tables should not be repeated in text. Only the most important ones should be emphasized.

R: Tables were revised in the text.

  1. Authors state in their comments that 'Spearman correlation wasn’t mentioned in our study', however Lines 156 and 157 in the Manuscript state: ' Spearman’s correlation analysis was used to evaluate correlation tests between numerical variables'.

R: Thank you for your attention. This statement has been removed from the text. It was included here by mistake because it was included in the statistics of our other article.

Quality of English language remains improper. The Manuscript must be revised by the native speaker.

R: Language, spelling and grammatical errors were corrected by a different native speaker.
